# 5-HT2a receptor in mPFC influences context-guided reconsolidation of object memory in perirhinal cortex

**Juan Facundo Morici[1,2], Magdalena Miranda[2,3], Francisco Tomás Gallo[2,3], Belén Zanoni[2], Pedro Bekinschtein[2,3], Noelia V Weisstaub[1,2]***

[1]Departamento de Ciencias Fisiológicas, Instituto de Fisiología y Biofísica Bernardo Houssay, Facultad de Medicina, Universidad de Buenos Aires, CONICET, Buenos Aires, Argentina; [2]Instituto de Neurociencia Cognitiva y Translacional, Universidad Favaloro, INECO, CONICET, Buenos Aires, Argentina; [3]Instituto de Biologia Celular y Neurociencias, Universidad de Buenos Aires, CONICET, Buenos Aires, Argentina

**Abstract** Context-dependent memories may guide adaptive behavior relaying in previous experience while updating stored information through reconsolidation. Retrieval can be triggered by partial and shared cues. When the cue is presented, the most relevant memory should be updated. In a contextual version of the object recognition task, we examined the effect of medial PFC (mPFC) serotonin 2a receptor (5-HT2aR) blockade during retrieval in reconsolidation of competing objects memories. We found that mPFC 5-HT2aR controls retrieval and reconsolidation of object memories in the perirhinal cortex (PRH), but not in the dorsal hippocampus in rats. Also, reconsolidation of objects memories in PRH required a functional interaction between the ventral hippocampus and the mPFC. Our results indicate that in the presence of conflicting information at retrieval, mPFC 5-HT2aR may facilitate top-down context-guided control over PRH to control the behavioral response and object memory reconsolidation.
DOI: https://doi.org/10.7554/eLife.33746.001

**\*For correspondence:**
noelia.weisstaub@gmail.com

**Competing interests:** The authors declare that no competing interests exist.

## Introduction

Flexibility and updating of memories is an adaptive capacity present in human and non-human animals. In rodents, fear and associative conditioning has been used as a way to study the underlying circuits and molecular mechanisms of flexible memories through analysis of extinction, reversal learning and updating of contingencies (*Lee et al., 2017*; *Rodriguez-Ortiz and Bermúdez-Rattoni, 2017*). However, little is known about the flexibility of episodic types of memory in spite of the evidence that the very process of remembering can affect memory stability. In particular, context-dependent memories might be an aid to guide adaptive behavior relaying on previous experience. For example, we would need to update memories from a previously dangerous neighborhood that is now a commercial district to adapt our behaviors to this new environment. Reconsolidation is the process by which consolidated memories enter a new period of instability after reactivation (*Miranda and Bekinschtein, 2018*; *Nader et al., 2000*) that can serve stabilization, strengthening, and updating (*Lee et al., 2017*; *McKenzie and Eichenbaum, 2011*; *Rodriguez-Ortiz and Bermúdez-Rattoni, 2017*). Its molecular mechanisms have been extensively studied in animals using fear and appetitive conditioning and spatial learning tasks (*Nader, 2015*). However, very little is known about the molecular and neural mechanisms of episodic memory reconsolidation, consistently observed in human autobiographical memory (*Forcato et al., 2007*; *Walker et al., 2003*; *Wymbs et al., 2016*). Retrieval of similar, but distinctive, experiences can be triggered by similar cues, so how the brain controls which memories are to be updated by reconsolidation is a key

question regarding the persistence of episodic memories. Here, we use object recognition memory in rodents as an episodic-like memory to tackle this important question. These elaborate memories involve multiple sensory pathways converging in multisensory areas, including the medial temporal lobe (MTL), of specific importance to episodic memory in humans. Interactions between PFC and hippocampus and other MTL structures may support the ability to create contextual representations, and use these contextual representations to retrieve the appropriate memories within a given context (*Preston and Eichenbaum, 2013*). Also neurons from the MTL respond to specific objects and contexts, suggesting that different areas may interact and work together to supportepisodic memory (*Ahn and Lee, 2017*; *Brandman and Peelen, 2017*; *Davachi, 2006*; *Diana et al., 2007*); *Lee and Park, 2013c*; *Preston and Eichenbaum, 2013*; *Squire et al., 2007*). Evidence indicates that different MTL regions store parts of the representation of a particular episode, and this raises the question of how this information is integrated, consolidated, retrieved, reactivated and reconsolidated. In addition, it is not known what happens after the memory of a given episode is reactivated; does the memory reconsolidate in all structures? Are different features of the memory reactivated in different regions? Is reactivation controlled by the same mechanisms in each region?

The serotonergic system is a key neuromodulator of PFC function and its relevance in memory processing has recently started to be understood (*Meneses, 2015*). The 5HT2a receptor (5-HT2aR) is one of the main post-synaptic serotoninergic receptor types and it is highly expressed in the PFC. A handful of studies suggest a role for 5-HT2aR in episodic memories in humans (*de Quervain et al., 2003*; *Wagner et al., 2008*), and a few animal studies have addressed a potential role of 5-HT2aR during memory consolidation (*Meneses, 2007*; *Meneses and Hong, 1997*; *Wagner et al., 2008*). We have previously shown that mPFC 5-HT2aR plays a role in recognition memory retrieval (*Bekinschtein et al., 2013*). Specifically, we demonstrated that 5-HT2aR activation is required when previously familiar and competing information is simultaneously presented. Thus, the activity of this receptor in mPFC could affect memory reactivation in downstream structures of the MTL. In this work we evaluate the importance of 5-HT2aR activation in the PFC in the control of retrieval and reconsolidation in different structures of the MTL when these processes are guided by contextual cues, similar to that which may occur during episodic memory retrieval in humans. We hypothesize that 5-HT2aR-mediated signaling in PFC during episodic memory retrieval differentially affects object memory reconsolidation depending on the contextual cues, and in that manner produces long lasting changes in episodic memory storage. Revealing the mechanisms of episodic memory retrieval and reconsolidation would be important to understand the function of these types of memory in guiding appropriate behavior in particular environments.

## Results

### 5-HT2aR activity in the mPFC modulates recognition memory reconsolidation in the PRH

Episodic memories are characterized for the encoding and retrieval of unique episodes. In rodents, episodic-like memories can be analyzed using a variety of behavioral paradigms. The Object in Context (OIC) task is a modified version of the spontaneous object recognition task characterized by having a strong spatial component, and can be considered an episodic-like task in rodents (*Aggleton et al., 1989* ; *Morici et al., 2015a*). To solve this task, animals require to retrieve a unique experience of an object explored in a particular context. As in other recognition memory tasks, it involves multiple sensory pathways and structures (*Preston and Eichenbaum, 2013*; *Warburton and Brown, 2015*). In particular, the PRH is engaged by specific object stimuli and signals the familiarity of those items, whereas the hippocampal region is involved in processing the contexts in which these events occurred (*Brown et al., 2010*). We used the OIC task to evaluate the mechanisms involved in control of retrieval and reconsolidation by the mPFC (*Figure 1a*). This task is a three-trial procedure that allows evaluation of the congruency between the context and the objects presented within it (*Wilson et al., 2013*). During the training phase, animals learned two different context-object associations. During test 1, the retrieval phase, a copy of each object used before is presented in one of the contexts. Thus, one of the objects is presented in an 'incongruent' context (object B = incongruent object), while the other is presented in a 'congruent' one (object A = congruent object). Novelty comes from a novel combination of an object and a context, and exploratory behavior will be driven

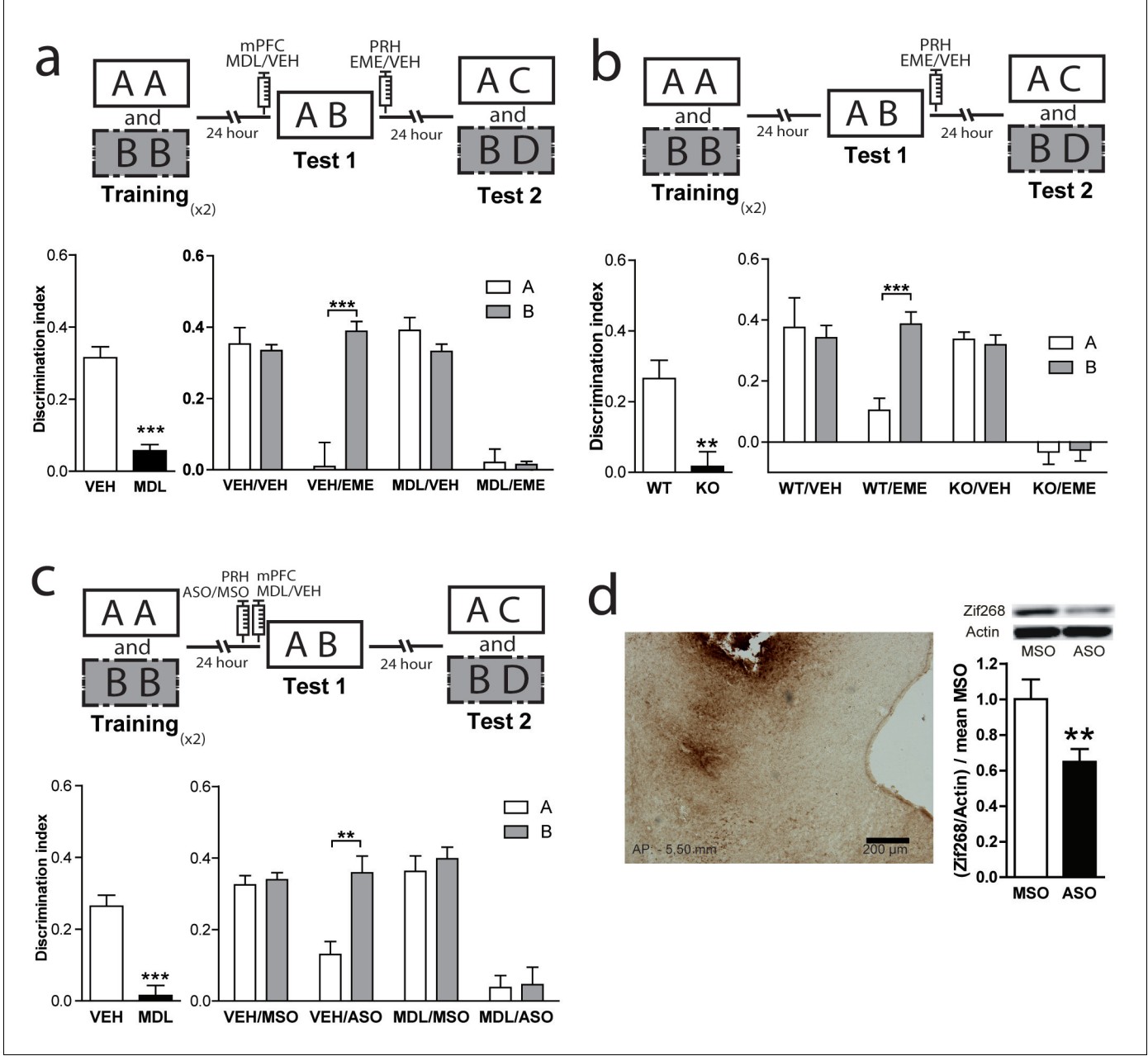

**Figure 1.** Reconsolidation of OIC memory traces in PRH is susceptible to manipulation of 5-HT2aR. (a) mPFC 5-HT2aR blockade alters object reconsolidation in PRH. Upper: Schematic representation of infusions and OIC paradigm in rats. Lower panels: Discrimination indexes during test phases, 24 and 48 hrs after the training sessions. Left: Infusion of MDL in mPFC impairs discrimination between contextually congruent and incongruent objects during test 1. Unpaired Student's t test (n = 12 per group), ***p<0.0001. Right: Infusion of EME in PRH immediately after test affects memory retention evaluated 24 hrs later (test 2). Infusion of MDL in mPFC before test 1 modified the memories affected by EME in PRH. Two-way ANOVA followed by Bonferroni post-hoc test, ***p$_{interaction}$ <0.0001 (n = 6 per group). (b) Genetic deletion of 5-HT2aR influences reconsolidation in PRH. Upper: Schematic representation of infusions and OIC paradigm in mice. Lower panels: Discrimination indexes during test phases, 24 and 48 hrs after training sessions. Left: Discrimination index for Test 1. Genetic deletion of 5-HT2aR (KO) affects discrimination between contextually congruent and incongruent objects compared with wild type (WT) mice (*Morici et al., 2015b*). Unpaired Student's t test, **p=0.0023 (n = 12–14 per group). Right: Discrimination index for test 2. The constitutive and global absence of 5-HT2aR expression alters the memory traces affected by EME infusion in PRH. Two-way ANOVA followed by Bonferroni post-hoc test, **p$_{interaction}$ <0.0029 (n = 6–7 per group). (c) mPFC 5-HT2aR blockade affects Zif268-dependent reconsolidation pattern of object memories in PRH. Upper panel: Schematic representation of infusions (ODNs in PRH and MDL/VEH in mPFC) and OIC paradigm. Lower panels: Discrimination indexes during test phases, 24 and 48 hrs after training sessions. Left: Discrimination index obtained during test 1, ***p<0.0001, unpaired Student's t test (n = 13–14 per group). Right: Infusion of Zif268 ASO in PRH after test 1 impaired memory retention evaluated during test 2. Previous infusion of MDL in mPFC modified the pattern of memory impairment induced by infusion of Zif268-ASO in PRH,

*Figure 1 continued on next page*

Figure 1 continued

*$p_{interaction}$ = 0.049, Two-way ANOVA followed by Bonferroni post-hoc test (n = 6–7 per group). (d) Left: Representative spread of 1 nmol of biotinylated ODN, 90 min post injection in the PRH. Right: Zif268-ASO injection decreases Zif268 protein levels. Top: representative Zif268 immunoblots of PRH protein extracts prepared 90 min after ODN infusion; actin was used as a loading control. Bottom: Quantification of Zif268 immunoblots. Actin-normalized Zif268 protein levels are shown, relative to the mean of the Zif268-MSO in the PRH of non-trained animals. **p=0.0055, unpaired Student's t test (n = 4 per group); data represent mean ±SEM. A and B represent objects presented to the animals in each session. During test sessions, A represents the contextually congruent and B the contextually incongruent objects.

DOI: https://doi.org/10.7554/eLife.33746.002

The following figure supplements are available for figure 1:

**Figure supplement 1.** Schematic representation of infusion site and coronal section showing the trace of the cannula.

DOI: https://doi.org/10.7554/eLife.33746.003

**Figure supplement 2.** Absolute exploration times reflect the amnesic effect of EME infusion after test 1.

DOI: https://doi.org/10.7554/eLife.33746.004

by retrieval of a particular 'what' and 'which context' conjunctive representation (*Eacott and Norman, 2004*). We have previously shown that blockade of mPFC 5-HT2aR signaling affects retrieval in the OIC, but has no effect in a one-item object recognition task (SNOR) (*Bekinschtein et al., 2013*), suggesting that blockade of mPFC 5-HT2aR selectively affects discrimination based on the combination of object identity and other features such as the context or the recency of the experience. In fact, after blockade of 5-HT2aR in mPFC during the test session, animals explore the congruent and incongruent object equally as if they are both equally familiar compared with a complete novel object (*Bekinschtein et al., 2013*). As memory retrieval can lead to reactivation and trigger reconsolidation, we hypothesized that 5-HT2aR inactivation might affect the pattern of object memory reconsolidation in PRH (for review [*Buckley, 2005*; *Bussey et al., 2005*; *Horne et al., 2010*]). After administration of MDL into the mPFC, the congruent and incongruent objects seemed to be retrieved with equal strength. Thus, we predicted that 5-HT2aR blockade in mPFC would render the incongruent and congruent object memory traces sensitive to protein synthesis inhibitors, such as emetine (EME), whereas in control animals, only the memory of the congruent object more strongly retrieved would be sensitive to the drug. Similar to what we described for short-term memories (*Bekinschtein et al., 2013*), infusion of the 5-HT2aR specific antagonist MDL 11,939 (MDL) in the mPFC impaired the resolution of OIC during test 1, 24 hr after acquisition (*Figure 1a*, left panel). MDL effect was transient as it did not affect the expression of the original object memories during test 2 in the MDL/VEH group (*Figure 1a*, right panel). Infusion of EME immediately after test 1 disrupted long-term memory retention during test 2 in the VEH/EME group for the congruent object (object A) only (*Supplementary file 1*). This suggests that the memory trace for the congruent object was labilized in PRH during test 1. However, blockade of mPFC 5-HT2aR during test 1 in combination with EME infusion into PRH immediately after test 1, disrupted long-term memory for both the incongruent (object B) and congruent (object A) objects, as observed in test 2 (*Figure 1a*, right panel and *Supplementary file 1*). This suggests that both memory traces were labilized becoming susceptible to the effects of EME (*Figure 1a*, right panel). Total training exploration time did not differ between objects A and B (paired t test, t = 0.078, p=0.9381, $mean_A$ = 48.99 ± 1.969, $mean_B$ = 49.16 ± 2.048). However, there was a difference in exploration time between the first and second exposures to each context during training (One-way ANOVA, F = 29,53, DF = 3, MS = 3968, $p_{main\ effect}$ <0.0001; significant Bonferroni multiples comparisons, context1 (Tr1 vs Tr2) $mean_{Tr1}$ = 30.08 ± 1.461, $mean_{Tr2}$ = 19.93 ± 1.173, p<0.0001, context2 (Tr1 vs Tr2) $mean_{Tr1}$ = 30.27 ± 1.728, $mean_{Tr2}$ = 17.44 ± . 9589, p<0.0001), that was expected, after re-exposure to the same context object association. EME affects the behavioral response of the rats during test 2. One conceivable possibility is that EME perceived the novel objects presented during test 2 (C or D) as familiar. Another possibility is that EME is blocking reconsolidation of the previously familiar objects, causing amnesia. If the second hypothesis is true, rats will lose the ability to remember the association between the objects and the previously familiar context showing enhanced exploration of object A. We compared the exploratory time of object A during test 2 between the VEH and EME groups. We found that animals infused with EME have enhanced exploration of object A relative to the VEH group, whereas there were no differences in the level of exploration of object B across groups (*Figure 1—figure supplement 2.*, panel a). Thus, animals infused with EME lost the memory for object A, they are not

perceiving the novel object as familiar. As pharmacological receptor antagonists could potentially bind to other targets, we confirmed the specificity of the function of 5-HT2aR signaling in OIC retrieval and reconsolidation using 5-HT2aR knockout (*Htr2a-/-*) mice. We cannulated *Htr2a-/-* and *Htr2a+/+* mice in PRH and trained them in the OIC task. We infused EME or VEH in the PRH immediately after test 1 (*Figure 1—figure supplement 1*), and analyzed retention of the original memories during test 2, (*Figure 1b*, upper panel). As previously reported (*Morici et al., 2015a*), *Htr2a-/-* mice had a deficit in the resolution of OIC task compared with *Htr2a+/+* mice in test 1 (*Figure 1b*, left panel and *Supplementary file 1*). Infusion of EME in the PRH immediately after test 1 impaired memory retention of the congruent object in *Htr2a+/+* mice and the retention of both the congruent and incongruent memory traces in *Htr2a-/-* during test 2 (*Figure 1b*, right panel and *Supplementary file 1*). This result suggests that both the memories for the congruent and incongruent objects undergo reconsolidation in the PRH of *Htr2a-/-*, but only the congruent object memory reconsolidates in the PRH of *Htr2a+/+* mice. Interestingly, the absence of 5-HT2aR did not affect the integrity of the original memory, as *Htr2a -/-* mice infused with VEH were able to recognize effectively each object during test 2, suggesting that mPFC 5-HT2aR activation is involved in the selection of the memory traces to be reactivated and not in the storage of the memories per se. Specificity of infusion site was controlled by infusion of EME after test 1, 0.1 mm above the area of interest. Under this condition, EME infusion had no affect on memory expression during test 2 in *Htr2a+/+* (Two-way ANOVA, F = 0.4167, $p_{main\ effect}$ = 0,5537, $DI_{ObjA-VEH}$ = 0.3754 ± 0.02858, $DI_{ObjA-EME}$ = 0.3753 ± 0.08477, $DI_{ObjB-VEH}$ = 0.4226 ± 0.06961, $DI_{ObjB-EME}$ = 0.3300 ± 0.04745, n = 3/group).

## Zif268 is necessary for OIC memory reconsolidation in the PRH

General protein synthesis inhibitors have been widely used in attempts to determine the reconsolidation process. However, under certain circumstances they can have other molecular targets not directly implicated in translation (*Alberini, 2008*). Zif268/Egr1 is a transcription factor important for reconsolidation of different types of memories including recognition memory (*Veyrac et al., 2014*), making it a good candidate to mediate reconsolidation in an OIC task. We investigated this hypothesis by infusing an antisense (ASO) or a scramble (miss-sense) control oligonucleotide (MSO) for Zif268 in the PRH in combination with pre-test 1 injection of MDL into the mPFC. Oligonucleotides are specific for target mRNAs and much less likely than protein synthesis inhibitors to affect other processes not related to translation. We hypothesized that if Zif268 was involved in the reconsolidation of OIC memory traces in PRH, the infusion of ASO would disrupt the reconsolidation of the memory traces labilized during test 1. During test 2, the VEH/ASO group showed amnesia only for the congruent object, while the MDL/ASO group showed amnesia for both the congruent and incongruent objects (*Figure 1d* and *Supplementary file 1*). The effect of ASO on test 2 cannot be explained by an effect of the ASO in memory retrieval during test 1, as in this instance, the VEH/ASO performance was similar to that of the VEH/MSO group (Unpaired t test, $mean_{VEH/MSO}$ = 0.2484 ± 0.04714, $mean_{VEH/ASO}$ = 0.2767 ± 0.04341, p=0.6664, t = 0.4430). This result indicates that expression of Zif268 is necessary for object memory reconsolidation in PRH. Immunodetection of a biotynilated oligonucleotide indicated that it did not spread beyond the PRH (*Figure 1d*, left panel). To ensure that Zif268-ASO was efficiently blocking protein expression, we infused Zif268-ASO or Zif268-MSO into the PRH 90 min before animals were sacrificed. Protein content was measure by western blot. Infusion of Zif268-ASO significantly decreased Zif268 protein levels in PRH (*Figure 1d*, right panel).

## Boundary conditions for reconsolidation in the OIC task in PRH

Different studies have described the requirements needed for reconsolidation of object recognition memory (*Akirav and Maroun, 2006*; *Balderas et al., 2015*; *Bozon et al., 2003*; *Clarke et al., 2010*; *Furini et al., 2015*; *Radiske et al., 2017*; *Rossato et al., 2007*; *Rossato et al., 2015*; *Sachser et al., 2016*; *Winters et al., 2011*). However, as far as we know, no other study has evaluated reconsolidation in the OIC task. Thus, we wanted to ensure that important reconsolidation hallmarks observed in other object recognition tasks were also present in the OIC task. We evaluated two characteristics highly relevant to the process we were studying, novelty during the retrieval session and effective memory reactivation during retrieval. In the first experiment, rats were trained as before, but test 1 consisted of exposure to two copies of a congruent object (*Figure 2a*). Immediately after, half of the

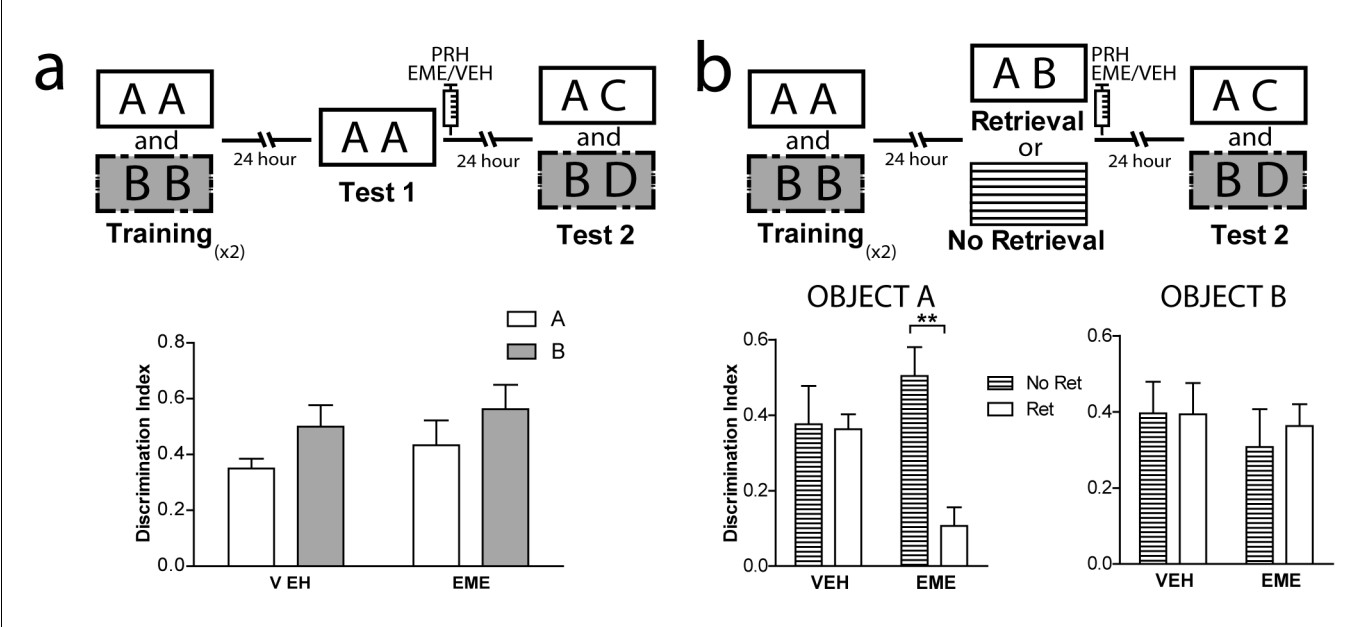

**Figure 2.** Boundary conditions for reconsolidation in the OIC task. (**a**) Upper panel: Schematic representation of infusions and behavioral paradigm. Rats were trained in the OIC paradigm but during test 1 were exposed to two copies of the contextually congruent object A. Immediately after test 1, rats were infused with VEH or EME in PRH and then exposed to the rest of the OIC task. Lower panel: Infusion of EME in PRH immediately after test 1 did not affect memory retention for any of the original objects. (n = 5 per group). (**b**) Upper panel: Schematic representation of infusions and behavioral paradigm. Rats were trained in the OIC paradigm. 24 hrs later, half the animals were exposed to a copy of the previously presented objects (A and B) and the other half were returned to their homecage. 24 hrs later memory retention was tested. Panel: Discrimination index for test 2 for object A. *$p_{interactionA}$ <0.0216, Two-way ANOVA followed by Bonferroni post-hoc test. Right: Discrimination index for test 2 for object B (n = 5 per group). Horizontal stripes represent no re-exposure group. Data represent mean ±SEM. A and B represent objects presented to the animals in each session. During test sessions, A represents the contextually congruent and B the contextually incongruent objects.
DOI: https://doi.org/10.7554/eLife.33746.005

animals were injected with EME and half with VEH into PRH. Both groups exhibited comparable memory retention for the congruent object during test 2 (*Figure 2a* and *Supplementary file 1*), indicating that a contextual mismatch between what has been experienced during training and what is present during test 1 is required for labilization/reconsolidation of object memories (*Hupbach et al., 2008*; *Pedreira et al., 2004*; *Sevenster et al., 2014*). In a second experiment, we compared memory retention during test 2 in rats that were exposed to the OIC paradigm (reactivation condition) and rats that at the time of the test 1 were returned to their homecages and thus memory retrieval was not engaged (homecage condition). Immediately after test 1 (for the reactivation condition) or 24 hr after training (for the homecage condition), the animals were infused with EME or VEH into PRH (*Figure 2b*). Only the animals from the reactivated group infused with EME showed significantly lower memory retention for the congruent object during test 2 (*Figure 2b* left panel and *Supplementary file 1*). All the other groups showed expected discrimination scores (*Figure 2b*, right panel). This result indicates that novelty during memory retrieval during test 1 is a necessary step for reconsolidation of the OIC memory and that object memories that are not retrieved are not susceptible to the action of protein synthesis inhibitors.

## Retrieval and reconsolidation are driven by contextual information in the OIC task

The correct resolution of the OIC task requires that animals recognize a novel combination of familiar objects and contexts. Context-guided retrieval is one of the main characteristics of episodic memory. However, little is known about the consequences of contextual reactivation on memory reconsolidation across different structures. If retrieval of object memories in the OIC task is context-guided, this task would be extremely relevant for the study of episodic memory reconsolidation. Thus, we propose that contextual information is an important cue for reactivation of relevant object

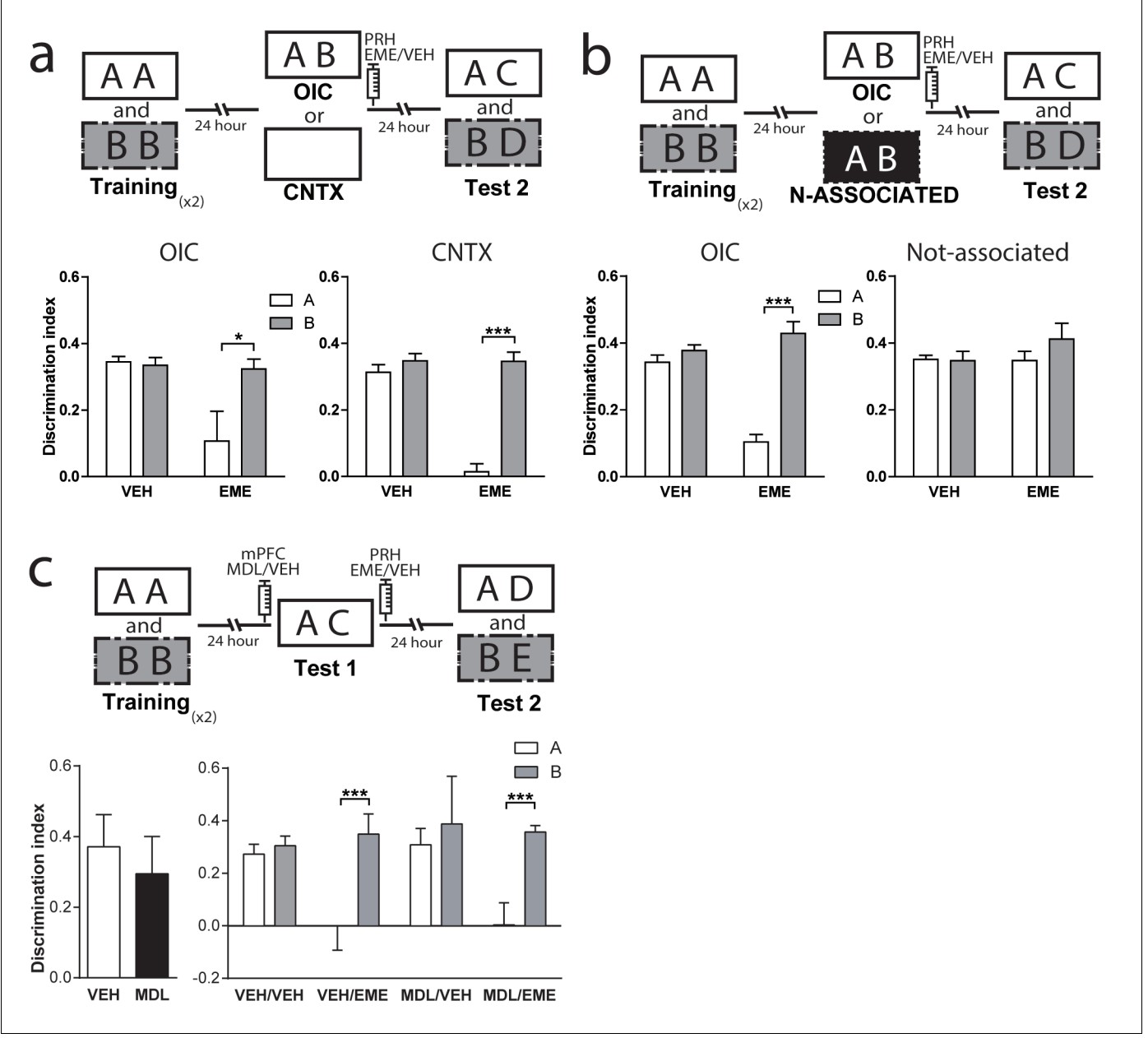

**Figure 3.** Contextual information drives object memory reconsolidation in PRH. (**a**) Schematic representation of infusions and behavioral paradigm. Rats were trained in the OIC task. For test 1, half the animals continued with the OIC protocol. The other half were exposed to one of the contexts alone. Immediately after test 1, rats were infused with VEH or EME in PRH. During test 2, memories of the original objects (A and B) were evaluated. Lower panels: Discrimination index for test 2. Left: OIC group. *$p_{drug}$ = 0.0197, $p_{interaction}$ = 0.0593, Two-way ANOVA followed by Bonferroni post-hoc test. Right: Context-only group. ***$p_{interaction}$ = 0.0003, Two-way ANOVA followed by Bonferroni post-hoc test (n = 6 per group). (**b**) Schematic representation of infusions and behavioral paradigm. Animals were trained in the OIC task and then divided into two groups. Half of the animals continued with the OIC task. The other half were given copies of objects A and B presented in a familiar way, but had not been previously associated with any of them. Immediately after test 1, rats were infused with VEH or EME in PRH. The black rectangle indicates a previously familiarized context that was not used during the training session, so not associated with the objects. Lower panels Discrimination index for test 2. Left: OIC group. ***p=0.0006, Two-way ANOVA followed by Bonferroni post-hoc test. Right: Not-associated group (n = 6 per group). (**c**) Upper: Schematic representation of infusions and behavioral paradigm. Rats were trained in the OIC task but during test 1 the contextually incongruent object was replace by a completely novel object transforming the paradigm into a SNOR task. Infusions into mPFC (VEH or MDL) and PRH (VEH or EME) were performed as indicated in the scheme. Lower panels: Discrimination index for test sessions. Left: Discrimination index for test 1. Right: Discrimination index for test 2. *** $p_{interaction}$ <0.0001, (n = 4–6 per group), Two-way ANOVA followed by Bonferroni post-hoc test. Data represent mean ±SEM. A and B represent objects presented to the animals in each session. During test sessions, A represents the contextually congruent and B the contextually incongruent objects.

DOI: https://doi.org/10.7554/eLife.33746.006

memories in the PRH. To test this hypothesis, animals were trained in the OIC task but during test 1 they were exposed to a familiar empty context in which objects A or B were experienced during training. We found that infusion of EME into PRH immediately after exposure to the context alone (test 1) was enough to cause amnesia for the contextually congruent object (*Figure 3a*, right panel and *Supplementary file 1*), but not for the incongruent object in test 2. This result suggests that the context cue can drive the labilization of the memory for objects previously associated with it, and that retrieval and reactivation of object memories in PRH is context-guided. If contextual information is used to guide retrieval, then presentation of the objects in a context not experienced before should not be sufficient for reactivation of object memories in PRH. To test this hypothesis, we first habituated rats to a third context different from the other two (trained contexts) and trained them in the OIC as before. During test 1, we presented a copy of each of object A and object B in this third familiar context in which no objects were experienced. Immediately after test 1, we infused EME in the PRH and 24 hr later we evaluated the memory for both objects in test 2. As predicted, infusion of EME into the PRH after test 1 did not produce any deficit in memory retention for any of the objects during test 2 (*Figure 3b* and *Supplementary file 1*). Thus, presentation of both objects in a context not associated with them was not enough for reconsolidation of object memories in PRH.

As described before, blockade of mPFC 5-HT2aR impaired retrieval in the OIC task. This result could be caused, for example, by failure to process the context correctly, that is the animals could be misremembering having seen the incongruent object in the context used for test 1. If this were true, infusion of MDL would allow reactivation and reconsolidation of the incongruent object even if it was absent during test 1. Thus, we designed a similar experiment in which animals were trained as described before but the incongruent object was swapped for a novel object during test 1 (*Figure 3c*). We found that both groups, independently of the manipulation of the mPFC spent more time exploring the novel object compared with the congruent object during test 1 (*Figure 3c*, left panel). Infusion of EME immediately after test 1 blocked the reconsolidation of the congruent trace independently of the infusion of MDL or vehicle in the mPFC (*Figure 3c*, right panel and *Supplementary file 1*), but had no effect on the reconsolidation of the incongruent object. This result suggests that in the absence of the incongruent object, serotoninergic modulation of mPFC 5-HT2aR was not necessary to solve the task. mPFC 5-HT2aR signaling might have a selective role in conditions in which correct retrieval is the consequence of recognizing a particular combination of an object and a context.

## 5-HT2aR blockade in the mPFC has no effect on reconsolidation of the OIC memory traces in the dorsal hippocampus

Classically, the hippocampus has been involved in different memory tasks that depend on contextual information (*Eichenbaum, 2017c*; *Izquierdo et al., 2016*; *Warburton and Brown, 2015*). Pharmacological studies indicate that the functionality of the dorsal hippocampus (dHPC) is required for the correct resolution of the OIC task (*Bekinschtein et al., 2013*). Also, protein synthesis in the CA1 region of the hippocampus has been shown to be necessary for object recognition memory reconsolidation (*Balderas et al., 2015*; *Clarke et al., 2010*; *Radiske et al., 2017*; *Rossato et al., 2015*; *Sachser et al., 2016*; *Winters et al., 2011*). Thus, we hypothesized that 5-HT2aR activation could modulate reconsolidation in the dorsal CA1 region of the HPC (dCA1), as it did in PRH. We bilaterally implanted cannulae in mPFC and dCA1. As expected, MDL infusion into mPFC before test 1 impaired performance in the OIC task (*Figure 4a*, left panel). However, blockade of dCA1 protein synthesis after test 1 affected memory retention for the incongruent object without impairing the retention of the congruent object. This effect was independent of the blockade of mPFC 5-HT2aR. Unexpectedly, our result suggests that the incongruent, but not the congruent, trace is reactivated in the dCA1 during test 1 in a mPFC 5-HT2aR-independent way (*Figure 4a*, right panel and *Supplementary file 1*). Similar to that observed for the infusion of EME in the PRH, the infusion of this drug in the dHPC appears to affect the reconsolidation; however, in this case, the reconsolidation of the incongruent object (B). If, as we predicted, the effect of EME is the blockade of reconsolidation, then rats would lose the ability to remember the association between the objects and the previously familiar context, showing enhanced exploration of object B. We analyzed the exploratory time between VEH and EME groups for objects A and B. We found enhanced exploration of object B in the EME group compared with the VEH group, indicating that EME-infused animals had effectively forgotten object B and not that they were treating the novel object as familiar. We also

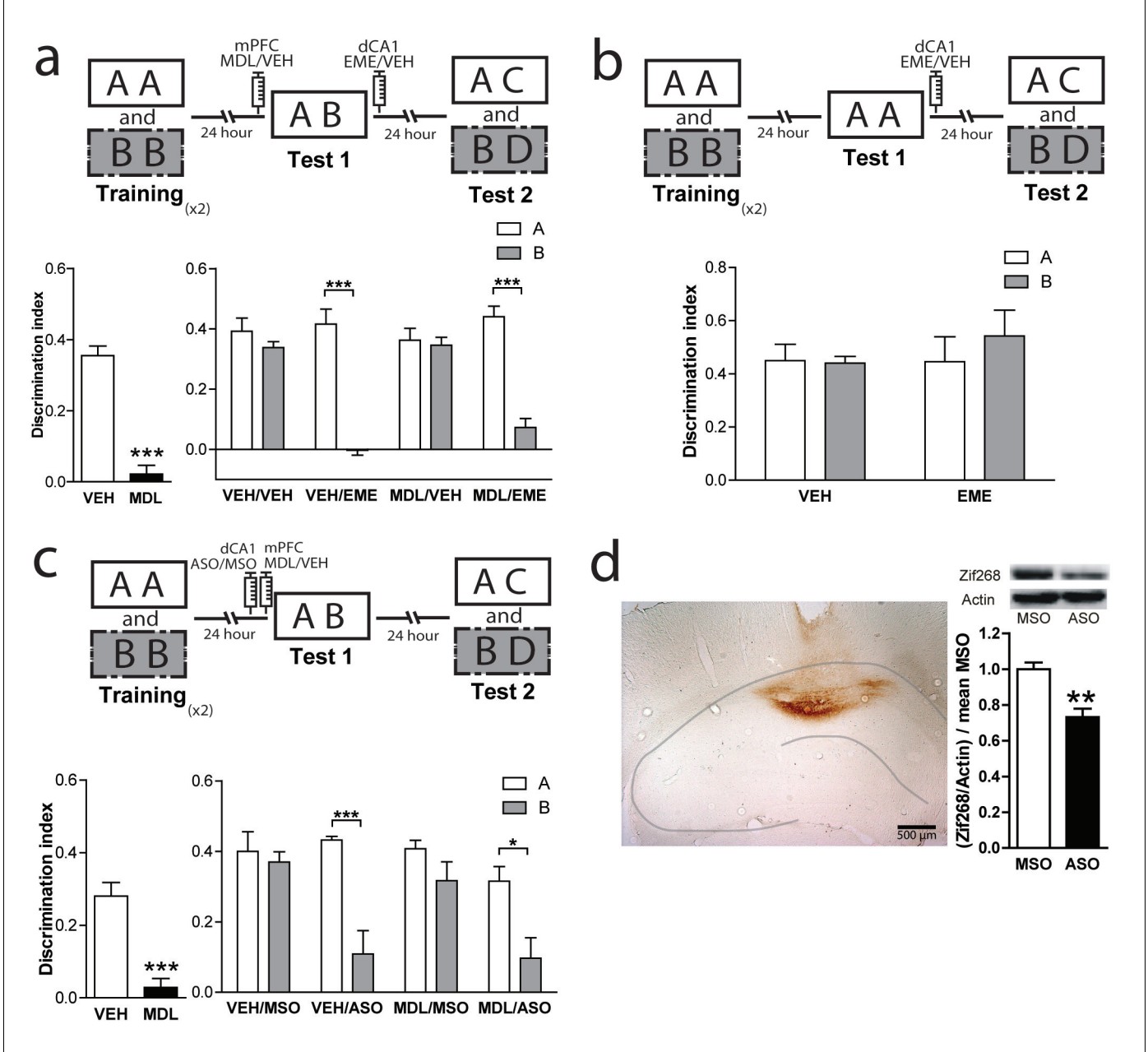

**Figure 4.** mPFC 5-HT2aR blockade has no effect on reconsolidation of object memory traces in the dCA1. (a) Upper panel: Schematic representation of infusions and OIC paradigm in rats. Lower panels: Discrimination indexes during test phases, 24 and 48 hrs after the training sessions. Lower left: Infusion of MDL in mPFC impairs discrimination between contextually congruent and incongruent objects during the test 1. ***p<0.0001, (n = 13–14 per group) unpaired Student's t test. Lower right: Infusion of EME in dCA1 immediately after test 1 affects memory retention evaluated 24 hrs later (test 2). Infusion of MDL in mPFC before test 1 did not modify the pattern of memories affected by EME. ***$p_{interaction}$ <0.0001, (n = 6–7 per group), Two-way ANOVA followed by Bonferroni post-hoc test. (b) Upper panel: Schematic representation of infusions and behavioral paradigm. Rats were trained in the OIC paradigm but during test 1 were exposed to two copies of the contextually congruent object A. Immediately after test 1, rats were infused with VEH or EME in dCA1 and then exposed to the rest of the OIC task. Lower panel: Infusion of EME in dCA1 immediately after test 1 did not affect memory retention for any of the original objects (n = 5 per group). (c) mPFC 5-HT2aR blockade does not affect Zif268-dependent reconsolidation pattern of object memories in the dCA1. Upper panel: Schematic representation of infusions (ODNs in PRH and MDL/VEH in mPFC) and OIC paradigm. Lower panels: Discrimination indexes during test phases, 24 and 48 hrs after training sessions. Left: Discrimination index obtained during test 1 ***p<0.0001, (n = 14 per group), unpaired Student's t test. Right: Discrimination index obtained during test 2. *$p_{interaction}$ = 0.03, ***p<0.0001, (n = 7 per group), Two-way ANOVA followed by Bonferroni post-hoc test. (d) Left: Representative spread of 1 nmol of biotinylated ODN, 90 min post injection in the dCA1. Right: Zif268-ASO injection decreases Zif268 protein levels. Top: representative Zif268 immunoblots of dCA1 protein extracts prepared 90 min after ODN infusion; actin was used as a loading control. Bottom: Quantification of Zif268 immunoblots. Actin-normalized Zif268 protein levels are

*Figure 4 continued on next page*

*Figure 4 continued*

shown, relative to the mean of the Zif268-MSO in the dCA1 of non-trained animals. **p=0.0019, (n = 5 per group), unpaired Student's t test. Data represent mean ±SEM. A and B represent objects presented in each session. During test sessions, A represents the contextually congruent and B the contextually incongruent objects.

DOI: https://doi.org/10.7554/eLife.33746.007

evaluated some of the conditions required for OIC reconsolidation in dHPC. We trained animals in the OIC task and, during test 1, we exposed the rats to two copies of the congruent object and infused EME or VEH in dCA1 immediately after. We observed that the infusion of EME had no effect on reconsolidation of the congruent or the incongruent objects, suggesting that novelty is also needed for reconsolidation in the dHPC (*Figure 4b* and *Supplementary file 1*).

## Zif268 is also required for OIC memory reconsolidation in the dCA1

As Zif268 is required in PRH for reconsolidation of OIC traces, we decided to evaluate its role on object memory reconsolidation in dCA1. Animals infused with ASO into the dCA1 before test 1 showed a significantly lower recognition of the incongruent object during test 2, but not of the congruent object compared with MSO-infused animals (*Figure 4c*, right panel and *Supplementary file 1*). Consistent with our findings using EME, amnesia was specific for the incongruent object and independent of the blockade of 5HT2aR in the mPFC. This result indicates that Zif268 participates in reconsolidation of the incongruent memory trace in dCA1. The effect of ASO on test 2 cannot be explained by any unspecific effect occurring during test 1, as the performance of the VEH/ASO and VEH/MSO groups was not different during this test (Unpaired t test, $mean_{VEH/MSO}$ = 0.3125 ± 0.06247, $mean_{VEH/ASO}$ = 0.2478 ± 0.04405, p=0.4136, t = 0.8470). We determined the area of infusion using a biotinylated oligonucleotide. Immunolabeling indicated that the oligonucleotide did not spread beyond the dCA1 (*Figure 4d*, left panel). To control that Zif268-ASO was efficiently blocking protein expression, animals were sacrificed 90 min after infusion of Zif268-ASO or Zif268-MSO. Protein content was measure by western blot. As expected, Zif268-ASO decreased Zif268 levels (*Figure 4d*, right panel).

## Interaction between mPFC and vCA1 is required for retrieval and reconsolidation of object memories in PRH

The resolution of OIC task might require contextual information used by the mPFC to control retrieval and reconsolidation of object memories. If contextual information is not stored in the mPFC, then it should be accessible to it from other areas such as the hippocampus. However, the ventral but not the dorsal hippocampus has direct unidirectional connections with mPFC (*Swanson et al., 1978*). Thus, we hypothesized that the pathway vHPC-mPFC could be important to provide contextually relevant information to the mPFC during test 1, required to solve the task (*Preston and Eichenbaum, 2013*). To test whether this direct connection played a role in the control of memory retrieval and reconsolidation in the OIC, we combined muscimol (MUS) or VEH infusions into the ventral CA1 (vCA1) right before test 1 with EME or VEH infusions into the PRH immediately after test 1 (*Figure 5a*). Silencing the vCA1 region during test 1 resulted in reconsolidation of the congruent and incongruent memory traces, while when the vCA1 region was active, only the congruent memory trace was sensitive to EME infusion. (*Figure 5a Supplementary file 1*). Importantly, if animals were trained in SNOR, which does not require integration of object and context, blockade of vCA1 activity had no effect on memory retention in test 2 (*Figure 5b*). This set of results indicates that vCA1 activity is necessary during retrieval for discrimination between the congruent and incongruent objects, but not for object recognition per se. As vCA1 sends direct, mostly ipsilateral, connections to the mPFC (for review [*Vertes, 2006*]), we postulated that both structures might be serially connected for the control of object-context memory retrieval and reconsolidation in the PRH. We performed a disconnection experiment on the vCA1 and mPFC circuit, combined with EME infusions into the PRH (*Figure 5c* and *Supplementary file 1*). We had two predictions: first, inactivation of vCA1 and blockade of 5-HT2aR in the mPFC ipsilaterally before test 1 should not produce any memory impairment during retrieval. On the other hand,

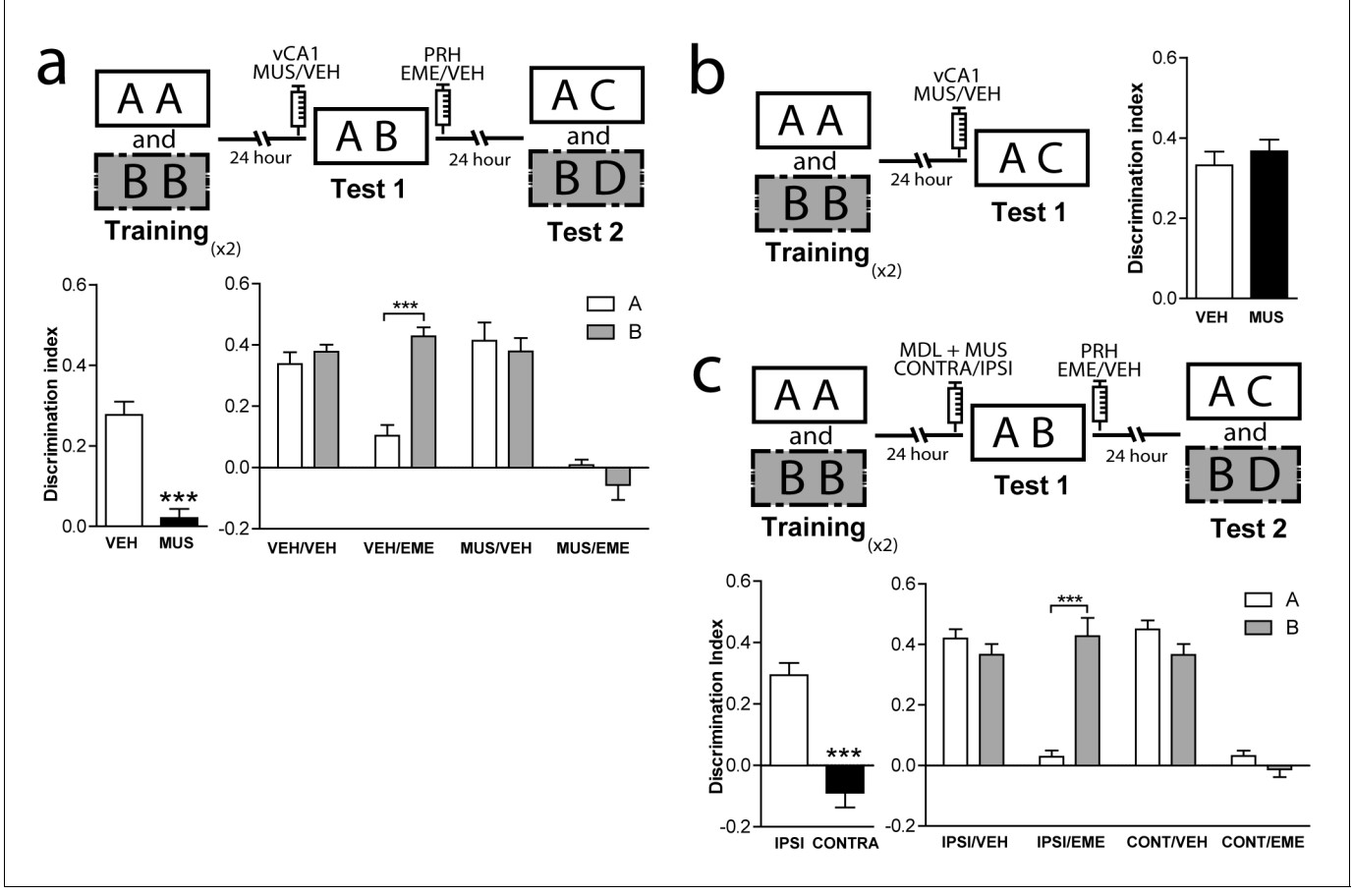

**Figure 5.** Interaction between mPFC and vCA1 is required for the control of memory retrieval and reconsolidation in the PRH. (**a**) Activity of vCA1 is necessary for the correct resolution of OIC task. Upper panel: Schematic representation of infusions and behavioral paradigm. Animals were trained in the OIC task and paradigm. Bilateral infusions of MUS were made in vCA1 before test 1. Emetine was infused in PRH immediately after test 1. Lower panels. Discrimination index obtained during test sessions. Left: Test 1. ***$p<0.0001$, (n = 13–15 per group), unpaired Student's t test. Right: Test 2. ***$p_{interaction}$ <0.0001, (n = 6–7 per group), Two-way ANOVA followed by Bonferroni post-hoc test. (**b**) Resolution of SNOR does not require vCA1 activity. Left panel: Schematic representation of infusions and SNOR version paradigm. Rats were trained in the SNOR task and infused with MUS immediately before test 1. Right panel: Discrimination index for test 1 (n = 5–6 per group). (**c**) vCA1 interacts with mPFC 5-HT2a to correctly solve the OIC task. Upper panel: Schematic representation of infusions and behavioral paradigm. Rats were trained in the OIC task, 15 min prior to test 1 MDL in the mPFC and MUS in vCA1 were infused in ipsi or contra-lateral hemispheres. For all the animals emetine was infused in PRH immediately after test 1. Lower panels: Discrimination indexes obtained during the test sessions. Left: Test 1. ***$p<0.0001$, (n = 16 per group), unpaired Student's t test. Right: Test 2. ***$p_{interaction}$ <0.0001, (n = 8 per group), Two-way ANOVA followed by Bonferroni post-hoc test. Data represent mean ±SEM. A and B represent objects presented to the animals in each session. During test sessions, A represents the contextually congruent and B the contextually incongruent objects.

DOI: https://doi.org/10.7554/eLife.33746.008

contralateral inhibition of these structures should cause memory impairment in test 1. Second, if vCA1 and mPFC were interacting during retrieval in test 1 to control memory reactivation and reconsolidation, ipsi or contralateral manipulation of the structures would produce distinct patterns of amnesia in PRH during test 2. Rats were infused ipsi- or contralaterally with MDL in the mPFC and MUS in the vCA1 immediately before test 1. All animals were bilaterally infused with EME in the PRH immediately after test 1. We found that ipsilateral blockade of 5-HT2aR signaling and vCA1 activity had no effect on retrieval of OIC task during test 1. However, when the same manipulation was done contralaterally, the congruent and incongruent objects were explored equally, indicating a memory deficit (*Figure 5c*, left panel and *Supplementary file 1*). During test 2, the ipsilateral manipulation resulted in sensitivity to EME for the congruent object memory trace only,

while the contralateral manipulation resulted in EME-dependent amnesia for both the congruent and incongruent objects (*Figure 5c*, right panel and *Supplementary file 1*). This result is indistinguishable from that observed after bilateral inhibition of 5HT2aR in mPFC or bilateral inactivation of vCA1 (*Figures 1a* and *5a*). Thus, these observations suggest that activity in the vCA1 interacts with 5-HT2aR-dependent activity in the mPFC to control context-guided retrieval and reconsolidation of object memories in PRH.

## Discussion

This study examined the role of mPFC 5-HT2aR signaling during OIC retrieval and reconsolidation. Our main findings are that (1) OIC memories reconsolidate at least in two structures, PRH and dCA1; (2) context guides reconsolidation of object memories in PRH and this process is sensitive to mPFC 5-HT2aR modulation; (3) context-guided reconsolidation of object memories in dCA1 is independent of mPFC 5-HT2aR activation; (4) Zif268 is required for context-guided reconsolidation of object memories in both structures; and (5) functional interaction between vCA1 and mPFC is necessary during retrieval to control context-guided reconsolidation in PRH.

Reconsolidation is defined as a post-retrieval restabilization process that shares important characteristics with the stabilization of memories that occur shortly after acquisition.Reconsolidation has been proposed as a mechanism for memory updating (*Lee et al., 2017*). Although human studies have shown reconsolidation-related changes in declarative memories (*Forcato et al., 2007*; *Walker et al., 2003*; *Wymbs et al., 2016*), most non-human animal studies of reconsolidation have focused on emotional memories (for review [*Careaga et al., 2016*; *Kroes et al., 2016*; *Lee et al., 2017*]). Only recently have researchers started to analyze the characteristics of reconsolidation of recognition memories in animals. In the OIC task, we observed that the animals distributed their exploratory behavior toward the contextually incongruent object during test 1. This suggests that they can recognize the identity of the objects but also the context in which they were presented. These complex and probably simultaneous assessments might require the combined activity of different brain structures, including the PRH, HPC and PFC (*Arias et al., 2015*; *Bachevalier et al., 2015*; *Eichenbaum, 2017b*; *Kinnavane et al., 2017*; *Lee and Lee, 2013b*; *Preston and Eichenbaum, 2013*).

The PRH has been proposed as a region where complex objects representations are stored (*Albasser et al., 2010a*, *2010b*; *Olarte-Sánchez et al., 2015*; *Buckley, 2005*; *Bussey et al., 2005*; *Horne et al., 2010*; *Schultz et al., 2015*), and it is particularly relevant for familiarity discrimination (*Brown and Banks, 2015*). The HPC is required for storage and processing of spatial information, and is a key structure for integration of object-context information (*Eichenbaum et al., 2007*; *Aggleton et al., 2012*). We found that in the OIC task, reconsolidation occurs at least in the PRH and dCA1. However, the memory feature reactivated and reconsolidated in these structures, appears to be different. The memory of the contextually incongruent object is reconsolidated in the dHPC, suggesting that contextual novelty is sufficient to labilize the memory trace in this region. On the other hand, the simultaneous presentation of the familiar object or only the context in which it was presented is enough to labilize the congruent memory trace in the PRH, suggesting that this region can rely on contextual information to selectively reactivate a particular object memory trace associated with that context. The differences observed in the type of information reconsolidated in each structure indicate that different features of episodic memories might be updated through this process in different brain regions. These results are in agreement with previous work in which it was shown that the hippocampus responds to spatial cues (object B would trigger a contextual mismatch), while PRH responds to the single objects (*Aggleton et al., 2012*)

Many studies support a role for mPFC in memory through cognitive or strategic control over memory retrieval in other brain areas (*Hernandez et al., 2017*; *Preston and Eichenbaum, 2013*; *Rajasethupathy et al., 2015*). During episodic memory retrieval, contextual cues are important to establish the trace dominance in the face of competition for retrieval resources (*Eichenbaum, 2017a*; *Farovik et al., 2008*; *Lee and Lee, 2013a*; *Livne and Bar, 2016*; *Preston and Eichenbaum, 2013*). In the OIC task, the congruent memory appears to be the most relevant to solve the task while the incongruent memory trace could somehow interfere with retrieval by activating a different episode. Changes in the behavioral response observed after mPFC 5-HT2aR blockade suggest that this receptor could be involved in the selection of the

dominant trace during retrieval and this could, in turn, affect context-guided reconsolidation of object memory traces. Consistent with this hypothesis, we found that mPFC 5-HT2aR blockade affected the reconsolidation pattern in the PRH but not in the dCA1, suggesting that serotonin modulation of mPFC function can exert direct top-down control on memory retrieval and reconsolidation in PRH (*Figures 1* and *3*). Our results suggest that engagement of mPFC appears to be a selective mechanism that occurs only when previously known information (e.g. objects A and B) is simultaneously presented but does not participate if the retrieval cues are linked to a single episode, such as when only a novel object is presented. This suggests that the reported modulation of mPFC activity was specific to discrimination between a congruent and an incongruent OIC, but not between a familiar (congruent) and a novel object. Although we cannot conclude from this set of results that this occurs because of the presence of competing episodes, a prediction that could be tested in future is that as the similarity of the episodes increases, so does 5-HT2aR modulation of the mPFC during retrieval.

It has been previously shown that Zif268 is necessary during both memory consolidation (*Jones et al., 2001*) and reconsolidation of SNOR (*Bozon et al., 2003*) using mice that are genetically deficient for this protein. In this work we decided to test whether Zif268 was also involved in the context-guided reconsolidation of object memories in PRH and HPC. Blockade of Zif268 expression in PRH and HPC recapitulated the deficits observed in the same structures using a broad protein synthesis inhibitor, suggesting that Zif268 might be part of the main mechanism involved in reconsolidation within these structures. Interestingly, infusion of MDL in mPFC before memory reactivation allowed the contextually incongruent trace to also be susceptible to Zif268-ASO infusion in the PRH. Then, Zif268 appears to belong to the core transcription mechanism of reconsolidation. Also, to our knowledge, our work is the first to identify a role for Zif268 during reconsolidation in PRH. Our results suggest that a similar transcriptional mechanism mediates reconsolidation for the OIC task in the HPC and PRH.

The effect observed on reconsolidation is 5-HT2a mediated, as the pattern of object memory reconsolidation in H*tr2a-/-* mice was similar to that observed by acute blockade of mPFC 5-HT2aR in rats. This result indicates that if there were a functional compensatory mechanism, this would not be sufficient to reverse the retrieval/reconsolidation deficit in the OIC. Importantly, these experiments specify that our pharmacological approach selectively disrupts 5-HT2aR signaling. Diffuse serotonin axons innervate and modulate PFC activity through excitatory and inhibitory receptors throughout the laminar organization of the cortex (*Lesch and Waider, 2012*), generating complex modulation of PFC function. Slice electrophysiological experiments indicate that one possible mechanism by which this could occur is by modulating the gain of mPFC neurons. Interestingly, 5-HT2aR principal neurons in the cortex are predominantly colossal cells that project to other cortical structures (*Avesar and Gulledge, 2012*), suggesting that they might have a rapid and direct effect on downstream cortical regions. Although our experiments do not address how blocking 5-HT2aR might be affecting context-guided retrieval, we could hypothesize that blocking 5-HT2aR during retrieval might alter the gain modulation of the cells and also the oscillation activity of the system, altering the output to target structures such as the PRH. Independently of the mechanism involved, the blockade of one particular serotonergic receptor, the 5-HT2aR, is enough to change the ability of the mPFC to correctly control context-guided retrieval when conflicting information is presented.

To control context-guided retrieval in PRH, mPFC might require to access or encode contextual information. Recent evidence suggests there is a bidirectional flow of information between the mPFC and the HPC (*Eichenbaum, 2017b*), wherein the events that initiate prefrontal control over memory retrieval may arise from the ventral part of the HPC (*Preston and Eichenbaum, 2013*). According to this scenario, environmental cues that define the context are processed by the ventral HPC. This context-defining information could be sent via direct projections to mPFC where neuronal ensembles would develop distinct representations. When subjects subsequently experience the same context, contextual information presented during retrieval would activate a more specific ensemble in mPFC to select the appropriate context representations in the dorsal HPC, while also suppressing context-inappropriate memories.

This model predicts that mPFC requires information from vHPC when animals are exposed to contextual relevant cues to control memory retrieval. Our results indicate that vHPC interacts with mPFC to control memory context-guided retrieval and reconsolidation in PRH, suggesting that both

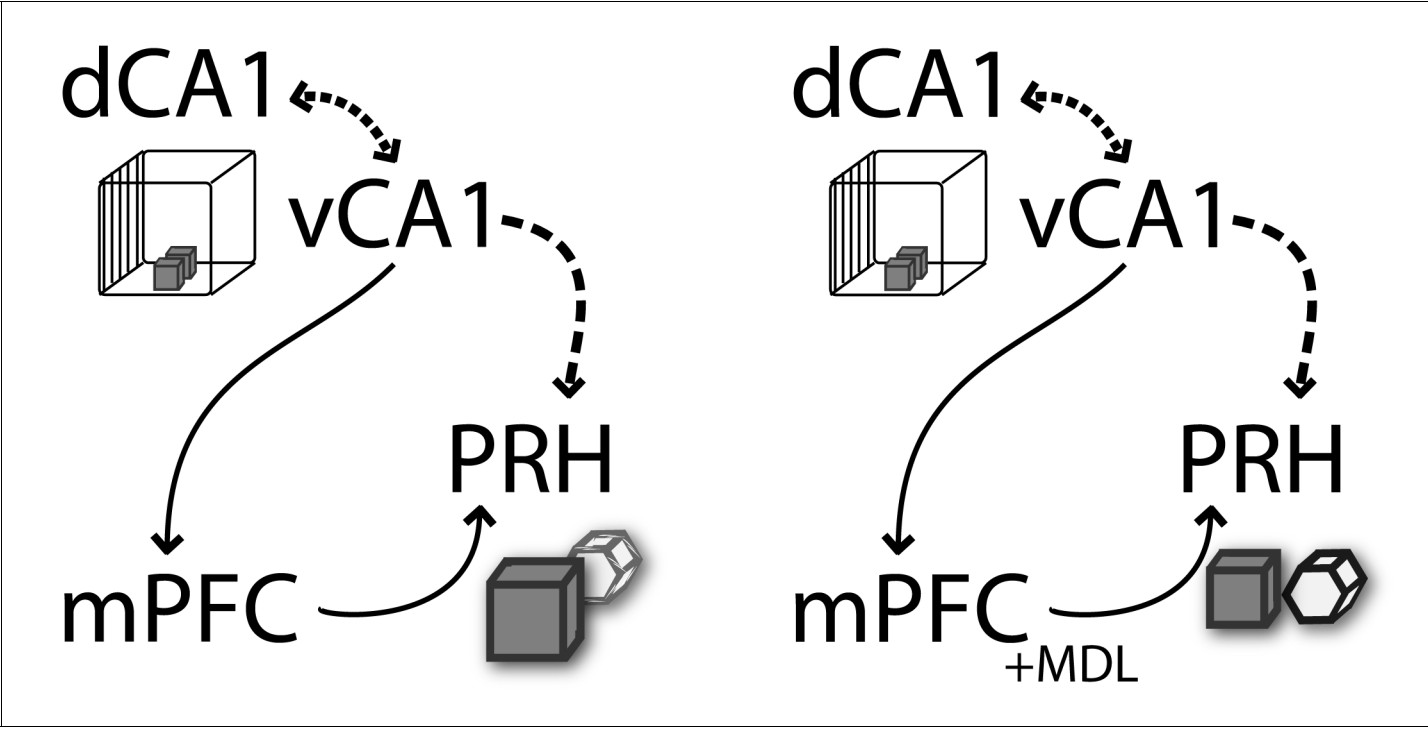

**Figure 6.** Schematic representation of functional connectivity of some of the structures involved in the retrieval and reconsolidation of OIC. mPFC and vCA1 interaction is required to control object memory trace retrieval and reconsolidation in PRH. The contextual information flows from the HPC (dCA1→vCA1) to mPFC. This information is used during retrieval by the mPFC to control the object memory traces expression and behavioral performance, as well as to influence the reconsolidation of object memories in the PRH. The arrows represent the flow of information suggested in this model. The discontinued lines represent the proposed interactions between structures (adapted from [*Preston and Eichenbaum, 2013*]).
DOI: https://doi.org/10.7554/eLife.33746.009

structures process information in series rather than as part of a parallel information circuit. Based on the direct unidirectional connection between vHPC and mPFC, we hypothesized that during test 1, vHPC provides contextual relevant information to the mPFC that is compared with previously acquired information to select the most relevant trace and control memory retrieval in PRH (*Figure 6*). Although our data do not support this directly, it is plausible that dHPC is the key area in which context and object information is assembled into a coherent trace. In our previous work (*Bekinschtein et al., 2013*), we showed that mPFC 5-HT2aR activity interacts with dHPC to solve the OIC task. Combining these data with our most recent findings, we can speculate that mPFC-dHPC interaction occurs through an indirect pathway involving PRH and vHPC. Thus, the contextual information that probably integrates spatial and object information is used to control memory reactivation in the PRH. When the animals were presented with information requiring integration of object and contextual features, mPFC would interact with PRH, controlling the strength of object memory expression. In the absence of 5-HT2aR signaling, the interaction between mPFC and PRH is severed and so is context-guided retrieval and reconsolidation in the PRH. This is observed as a deficit in behavioral discrimination as well as in the labilization of object traces in PRH. This lack of retrieval selectivity has a durable effect on long-term memory, leading to persistent changes of the original memory traces.

Overall, our study suggests that mPFC 5-HT2aR has been overlooked as an important player in control of episodic memory retrieval and in the circuitry involved in memory reconsolidation in downstream structures in the medial temporal lobe. As episodic memory deficits are one of the main cognitive characteristics observed across psychiatric disorders, 5-HT2aR modulation could be a

molecular target to improve cognitive deficits related to episodic memory retrieval and long-term expression.

# Materials and methods

## Key resources table

| Reagent type (species) or resource | Designation | Source or reference | Identifiers | Additional information |
|---|---|---|---|---|
| strain, strain background | Wistar rats, 129 Sv/Ev mice | Facultad de Ciencias Exactas y Naturales, UBA Animal Facility and in house animal facility for the mice. Mice are original from Taconic | | |
| antibody | zif268, actina | Santa Cruz | Sc189, sc1615 | |
| sequence-based reagent | Zif268, antisense | Sigma | 80103076030020 HAO2389671 | |
| commercial assay or kit | vectastin ABC kit | Vector lab | PK 6100 | |
| chemical compound, drug | MDL 11,939 | Tocris | #0870/10 | |
| chemical compound, drug | Emetine | Sigma | #7083-71-8 | |
| chemical compound, drug | Muscimol | Sigma | #2763-96-4 | |
| software, algorithm | GraphPad | GraphPad sofware | | |

## Ethic statement

All experimental procedures were in accordance with institutional regulations (Institutional Animal Care and Use Committee of the School of Medicine, University of Buenos Aires, ASP # 49527/15) and government regulations (SENASAARS617.2002). All efforts were made to minimize the number of animals used and their suffering

## Subjects

A total of 383 animals were used. Species and number of animals in each group are as follows.

332 Male adult Wistar rats

51 *Htr2a −/−* and *Htr2a+/+* male 129Sv/Ev mice. Strain details of these transgenic mice are described elsewhere (*Weisstaub et al., 2006*).

At the time of the experiments rats weighed between 200 and 250 gr and were housed in groups. Mice were 8–12 weeks old and bred in house. After weaning, male mice of both genotypes were housed five per cage. Mice and rats were kept with water and food *ad libitum* under a 12 hr light/dark cycle (lights on at 7:00 A.M.) at a constant temperature of 23°C. Experiments took place during the light phase of the cycle (between 10:00 A.M. and 5:00 P.M.) in quiet rooms with dim light.

## Surgery and drug infusions

Animals were deeply anesthetized with ketamine (Holliday, 80 mg kg$^{-1}$ i.p. for rats and 150 mg kg$^{-1}$ i.p. for mice) and xylazine (Konig, 8 mg kg$^{-1}$ i.p. for rats and 6.6 mg kg$^{-1}$ i.p. for mice) and placed in a stereotaxic frame (Stoelting). We list below the relevant stereotaxic coordinates for each structure and strain. The skull was exposed and adjusted to place bregma and lambda on the same horizontal plane. After small burr holes were drilled, one, two or three pairs of guide cannulae (22 g for rats and 23 g for mice) were implanted bilaterally into one or more structures described below.

After the surgery and until the infusions, a dummy injector (30 G) was inserted into the guide cannulae to preclude their obstruction

## Cannula placement

To check cannula placement, 24 hrs after the end of the behavioral experiments animals were infused with 1 µl of methylene blue through the dummy cannulae and 15 min later deeply anesthesized and sacrificed. Histological localization of the infusion sites were established using magnifying glasses. No animals were excluded because of cannulae misplacement. For analysis of ODN spread after injection, rats were injected with 2 nmol/µl (0.5 µl/side) of biotinylated ASO ODN, and 2 hr later, they were anesthetized and perfused transcardially with 0.9% saline followed by 4% paraformaldehyde. The brains were isolated and sliced, and the ASO was detected by avidin–biotin staining.

### Experiments made in rats

mPFC (anterior—posterior (AP) +3.20 mm / lateral (LL) ±0.75 mm / dorsoventral (DV) −3.50 mm; see *Figure 1a*).

PRH (AP −5.50 mm / LL ± 6.30 mm / DV −7.10 mm; see *Figure 1b*) dCA1 (AP −3.90 mm / LL ± 3.00 mm / DV −3.00 mm; see *Figure 1c*), vCA1 (AP −6.30 mm / LL ± 5.50 mm / DV −5.50 mm; see *Figure 1d*).

### Experiments in mice

PRH (AP −3.05 mm / LL ± 4.55 mm / DV −3.55 mm; see *Figure 1e*).

At the end of surgery, animals were injected with a single dose of meloxicam (0.2 mg/kg) as analgesic and gentamicin (0.6 mg/kg) as antibiotic. Behavioral procedures commenced 5–7 d after surgery.

On test 1, the dummy injector was removed and a 30 G injection cannula extending 1 mm below the guide cannula was inserted. The injection cannula was connected to a 10 µl Hamilton syringe. For the experiments made in rats, animals bilaterally received 1 µl infusion of the appropriate drug or its vehicle into the mPFC or the vCA1 15 min before test 1. In the mPFC we performed infusions of a 5-HT2aR antagonist MDL 11,939 (MDL, 300 ng/µl, 1 µl per side, Tocris Bioscience, Pittsburgh) or vehicle (5% DMSO in saline). In the vCA1, animals received infusions of the GABA$_A$ receptor agonist muscimol (MUS, 0.1 µg/µl, 1 µl per side, Sigma, St. Louis) or vehicle (saline). For the disconnection experiment (see *Figure 4B*), animals received contralateral or ipsilateral infusions of MDL in the mPFC and MUS in the vCA1 15 min before test 1. Depending on the experiment, animals received bilateral infusion of the protein synthesis inhibitor, emetine (EME, 50 µg/µl, 1 µl per side, Sigma) or vehicle (saline) immediately after test 1 in PRH or dCA1. In other experiments (see *Figures 2C* and *3C*), dCA1 or PRH bilateral infusions of zif268 antisense (ASO, 2 nmol/µl, 0.5 µl per side, Sigma, 5'GGT AGT TGT CCA TGG TGG-3) or missense oligonucleotides (MSO, 2 nmol/µl, 0.5 µl per side, Sigma, 5′-GTG TTC GGT AGG GTG TCA-3′) were used, as described in *Lee et al., 2004*. The oligodeoxynucleotides infusions were made 90 min before test 1 in the PRH or dCA1. The allocation of each animal in an experimental group was randomly assigned. After the behavioral procedures, rats received a bilateral infusion of methylene blue to confirm drug infusion sites (*Figures 1*, *2*, *3*, *4* and *5*, *Figure 1—figure supplement 1*).

## Behavioral experiments

### Video scoring

All experiments were recorded using Samsung HMX-F80 cameras. The cameras were located on top of each arena allowing visualization of the complete space. Offline analysis was performed using manual chronometers for all phases of each experiment by a trained experimenter who was blind to the conditions of the experiment.

### Apparatus

We used two different shaped mazes for the experiments with rats and mice. For the experiments done with rats, the first apparatus was a 50 cm wide X 50 cm length X 39 cm height arena with black plywood walls. The floor was black with white lines dividing it into nine squares. The second apparatus was a 60 cm wide X 40 cm length X 50 cm height acrylic rectangle. The floor was white as well as two of its walls, which had different visual clues. The frontal wall was transparent and the back wall was hatched (*Bekinschtein et al., 2013*). For experiments with mice we used a rectangular

shaped apparatus and a triangular shaped apparatus made of gray foam board. Both apparatus were 40 × 25 cm length ×30 cm high (*Morici et al., 2015b*). All contexts used within species had the same surface area to avoid differences resulting from the size of the arena. Duplicate copies of objects made from plastic, glass and aluminum were used. Objects were thoroughly cleaned between phases and randomly assigned to the different phases of the experiments. The heights of the objects ranged from 8 to 24 cm (depending on whether the experiment involved mice or rats) and they varied with respect to their visual and tactile qualities. All objects were affixed to the floor of the apparatus with an odorless reusable adhesive to prevent them being displaced during each session. The objects were always located along the central line of the maze, away from the walls and equidistant from each other. As far as we could determine, the objects had no natural relevance for the animal as they were never associated to any reinforcement. The objects, floor and walls were cleaned with ethanol 50% (for rats) or 10% (for mice) between experiments. Exploration of an object was defined as directing the nose to the object at a distance of <2 cm and/or touching it with the nose. Turning around or sitting on the object was not considered exploratory behavior (see *Video 1*).

## Object in Context task (OIC)

This task is a three-trial procedure that allows evaluation of the congruency between the context and the object (*Wilson et al., 2013*). During the training phase, animals are exposed to two different pairs of identical objects presented in two different contexts (see *Figure 1a*). These presentations are separated by 1 hr. During test 1, carried out24 hrs after the last presentation, a new copy of each of the objects used before is presented in one of the contexts (pseudorandomly assigned). Thus, one of the objects is presented in an 'incongruent' context (object B = incongruent object), while the other is presented in a 'congruent' one (object A = congruent object). In this task, novelty comes from a novel combination of an object and a context, and exploration will be driven by retrieval of a particular 'what' and 'which' context conjunctive representation (*Eacott and Norman, 2004*). We referred to the objects in the text as congruent and incongruent for descriptive purposes. In the figures congruent objects are referred to as 'A' and incongruent objects as 'B'.

### Habituation sessions

Rats were habituated to each context in which they were allowed to explore each context for 10 min. Mice were habituated during 6 consecutive days to increase their exploratory behavior. On the first day they spent 10 min in each arena. In the subsequent five habituation sessions, mice were exposed for 5 min/arena each time as previously described (*Morici et al., 2015b*).

### Training sessions

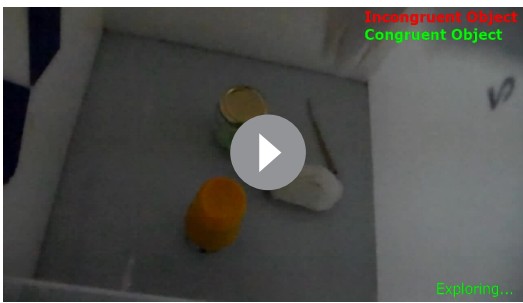

**Video 1.** Representative fragment of a test 1 session. The rat was exposed to copies of objects A and B and allowed to freely explore them. Exploration of the objects was defined as directing the nose to the object at a distance of <2 cm and/or touching it with the nose. Turning around or sitting on the object was not considered exploratory behavior.
DOI: https://doi.org/10.7554/eLife.33746.010

In the first training session, two identical objects (A1 and A2) were placed into one of the arenas (context 1). The animals were introduced into the context facing the wall and were then allowed to explore the environment for 5 min. At the end of this session, the animals were returned to their homecage. After a 30 min delay, animals were reintroduced to context 1, and they were allowed to explore the same objects for another 5 min, and then they were returned to their homecage. The same procedure was used to train the animals in context two using a second pair of objects (B1 and B2). The $Tr2_A$ was separated from $Tr1_B$ by 1 hr. The arenas were pseudorandomly assigned as context 1 or 2.

### Test 1

During this session animals were re-exposed to previously familiar contextual and object features

for 3 min. However, only one combination of objects-contexts will be shown in the paradigm schema to make easier the comprehension of the experimental design. For experiments described in *Figures 1*, *2a*, *3a, c*, *5a and c*, animals were reintroduced to context 1 or context 2 and were allowed to explore for 3 min one copy of object A and one copy of object B (OIC version). In *Figures 3c* and *5b*, animals were reintroduced to one of the contexts and were allowed to explore a copy of object A or object B (depending on the retrieval context) and one copy of a new object, spontaneous novel object recognition version (SNOR version). In experiment shown in *Figure 3b*, animals were exposed for 3 min to one copy of object A and one copy of object B in a third context, previously habituated but not used during the training session (not-associated version). For the experiment shown in *Figure 3a*, rats were exposed to context 1 or 2 but in this case without objects (context-only version) or with two copies of object A or B, depending on the retrieval context used (see *Figures 2a* and *4b*, no-novelty version). For the not-retrieved version rats were infused and returned directly to their homecage (see *Figure 2b*, not-retrieved version).

### Test 2

Each animal was re-exposed to context 1 and was allowed to explore for 3 min a copy of object A and a copy of a new object. After that, the animals were reintroduced to context 2 and were allowed to explore for 3 min a copy of object B and a copy of another new object.

## Behavioral analysis

For each behavioural session time exploring each object was measured. For sample and test phases, we analyzed the total exploration time for every copy of the objects in each session. For the test sessions we also calculated the Discrimination Index (DI). In test 1, the DI was calculated as $t_{incongruent} - t_{congruent}$/total exploration (time exploring the congruent +incongruent objects) of the session. In the particular case in which during test 1 animals were exposed to a novel object instead of an incongruent, then the DI was calculated as $t_{novel} - t_{congruent}$/total exploration In test 2 a DI for each of the trials was calculated as $t_{novel} - t_{familiar}$/total exploration.

## Criteria of exclusion

Animals that explored the objects for less than 5 s during any of the phases were excluded from the experiments. Once the animals recovered from surgery the behavioral procedure started. Rats were run in groups of 8–10 per week and randomly assigned to each experimental group at the beginning of the experiment. So, all conditions within an experiment were run simultaneously. For example, for experiment 1a rats were divided between the Veh/veh; Veh/Eme; MDL/Veh; MDL/EME. Sample size was determined. Overall, 29 rats and 18 mice were excluded based on this criterion (see *Supplementary file 2*).

As a result of the loss of 12 video recordings from the experiment presented in *Figure 3c*, we excluded those animals from all the data and analysis presented. One mouse was excluded from the analysis because its was H*tr2a*+/-.

Exploration times for test 1 or test 2 that were outside the confidence interval (mean ± 2 SD) were considered to be outliers and excluded from the sample. None of the animals were excluded based on this criterion.

## Immunoblot assays

PFC and HIP tissue punches were homogenized with a micropestle in an ice-chilled buffer (20 mM Tris-HCL [pH 7.4], 0.32 M sucrose, 1 mM EDTA, 1 mM EGTA, 1 mM PMSF, 10 µg/ml aprotinin, 15 µg/ml leupeptin, 10 µg/ml bacitracin, 10 µg/ml pepstatin, 15 µg/ml trypsin inhibitor, 50 mM NaF, and 1 mM sodium orthovanadate). Samples of homogenates (10 µg of protein) were subjected to SDS-PAGE (10% gels) under reducing conditions. Proteins were transferred onto PDVF membranes in transfer buffer (25 mM Tris, 192 mM glycine, 10% v/v methanol) for 16 hr at 40V and 4°C. Western blots were performed by incubating membranes first with anti-Zif268 antibody (C19, rabbit polyclonal IgG, 1:1000, Santa Cruz Biotechnology Inc, Santa Cruz, CA), then stripped and incubated with anti-actin antibody (C-11, goat polyclonal IgG, 1:10000, Santa Cruz Biotechnology Inc, Santa Cruz, CA). The ECF system was used for chemifluorescent immunodetection. Western blots were quantified using ImageJ software (Version 1.51 k, NIH).

## Statistical analysis

Statistical analyses were performed with GraphPad 6.01. Behavioral data were analyzed using two-tail unpaired Student's t test when two groups were compared. For comparisons between two repeated-measured groups, two-tail paired Student's t test was used. Some of the data shown in *Supplementary file 3* were transformed before the analysis. One- or Two-way ANOVA followed by Bonferroni post-test, as indicated in the figure legends, was used when three or more groups were involved. In all cases, p values were considered to be statistically significant when $p<0.05$. All data are presented as the mean ± s.e.m

As the behavioral data from training sessions of the experiments made in rats were similar across experiments, we ran a One-way ANOVA after pooling three trainings from all the experiments presented in this work. Total exploratory time during test session 1 is presented in *Supplementary file 3*. No significant differences in total exploration between treatments were observed for any of the experiments. Differences between exploratory time for objects A and B were observed in all vehicle- and in some of the MDL-treated groups across experiments (*Supplementary file 4*), suggesting that the experimental procedures had no deleterious effect on the subjects' exploratory behavior but it did affect their discrimination in a task-specific manner. Total exploratory time during test session 2 is presented in *Supplementary file 1*. No significant differences in total exploration for any of the sessions that conform to test 2 were observed for any of the experiments, with the exception of the EME group described in *Figure 5C*.

## Acknowledgement

We would like to thank Dr Lionel Muller Igaz, Dr Emiliano Merlo and Dr Juan Belforte for their critical reading and comments on the manuscript. We would like to thank Graciela Ortega, Veronica Risso and Jessica Unger for technical support. This work was supported by research grants from the National Agency of Scientific and Technological Promotion of Argentina (ANPCyT) to NVW (PICT 2012–0927 and PICT 2015–2344) and from the University of Buenos Aires to NVW (Ubacyt 2014–2017 GEF).

## Additional information

### Funding

| Funder | Grant reference number | Author |
|---|---|---|
| Agencia Nacional de Promoción Científica y Tecnológica | Recently formed grouped PICT D 2012-0927 | Noelia V Weisstaub |
| Agencia Nacional de Promoción Científica y Tecnológica | PICT A 2015-2344 | Noelia V Weisstaub |
| Universidad de Buenos Aires | Ubacyt 2014–2017 GEF | Noelia V Weisstaub |

The funders had no role in study design, data collection and interpretation, or the decision to submit the work for publication.

### Author contributions

Juan Facundo Morici, Data curation, Formal analysis, Investigation, Methodology, Writing—original draft, Writing—review and editing; Magdalena Miranda, Francisco Tomás Gallo, Belén Zanoni, Investigation, Methodology; Pedro Bekinschtein, Conceptualization, Formal analysis, Supervision, Writing—original draft, Writing—review and editing; Noelia V Weisstaub, Conceptualization, Formal analysis, Supervision, Funding acquisition, Writing—original draft, Project administration, Writing—review and editing

### Author ORCIDs

Juan Facundo Morici http://orcid.org/0000-0002-5207-0428
Magdalena Miranda http://orcid.org/0000-0002-6789-274X
Francisco Tomás Gallo http://orcid.org/0000-0003-1393-0218

Belén Zanoni [ID] https://orcid.org/0000-0002-8898-9911
Pedro Bekinschtein [ID] https://orcid.org/0000-0002-0004-9619
Noelia V Weisstaub [ID] http://orcid.org/0000-0003-3444-1915

## Ethics

Animal experimentation: All experimental procedures were in accordance with institutional regulations (Institutional Animal Care and Use Committee of the School of Medicine, University of Buenos Aires, ASP # 49527/15) and government regulations (SENASAARS617.2002). All efforts were made to minimize the number of animals used and their suffering

## Decision letter and Author response

Decision letter https://doi.org/10.7554/eLife.33746.017
Author response https://doi.org/10.7554/eLife.33746.018

## Additional files

### Supplementary files

• Supplementary file 1. Exploration time for each object presented during test 2. Mean ± SEM exploratory time for each object (A, B, C and D) presented during test 2. Raw exploratory times were analyzed separately by animals infused with VEH or MDL. Two-way ANOVA followed by Sidak's multiple comparisons were made, where one factor was the object (A vs. C and B vs. D) and the other factor was the treatment (e.g. emetine vs. vehicle). $\alpha < 0.05$, ****$p<0.0001$, ***$p<0.001$, **$p<0.01$, *$p<0.05$. For the experiment described in *Figure 3a* , the data were transformed because they did not follow a normal distribution.
DOI: https://doi.org/10.7554/eLife.33746.011

• Supplementary file 2. Animals excluded based on the exclusion criteria. Animals that explored any of the objects less than 5 s during any of test phases were excluded from the experiment.
DOI: https://doi.org/10.7554/eLife.33746.012

• Supplementary file 3. Total exploration time during test 1. Total exploratory time is presented for experiments in which a pharmacological or genetic manipulation is presented. No treatment effect is observed. Data represent mean ± SEM. Unpaired t test, $\alpha < 0,05$.
DOI: https://doi.org/10.7554/eLife.33746.013

• Supplementary file 4. Exploration time for congruent and incongruent objects during test 1. Mean ± SEM exploratory time. for each object during test 1. Object A represents the contextually congruent object; Object B represents the contextually incongruent object. Test 1. Two-way ANOVA with multiple comparisons, Sidak's multiple comparisons test $\alpha < 0.05$, ****$p<0.0001$, ***$p<0.001$, **$p<0.01$, *$p<0.05$.
DOI: https://doi.org/10.7554/eLife.33746.014

• Transparent reporting form
DOI: https://doi.org/10.7554/eLife.33746.015

### Data availability

All data generated or analysed during this study are included in the manuscript and supporting files.

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
