## [Decision Letter]

Thank you for submitting your article "5-HT2a receptor in mPFC controls context-guided retrieval and reconsolidation of object memory in Perirhinal Cortex" for consideration by *eLife*. Your article has been reviewed by three peer reviewers, and the evaluation has been overseen by a Reviewing Editor with Richard Ivry coordinating the process as the Senior Editor. The following individual involved in review of your submission has agreed to reveal their identity: Karim Nader (Reviewer #2).

The reviewers have discussed the reviews with one another and the Reviewing Editor has drafted this decision to help you prepare a revised submission.

Summary:

This paper presents results from a comprehensive set of experiments examining mechanisms of object-in-context memory reconsolidation. The key finding is that 5-HT2a receptors in the medial prefrontal cortex modulate reconsolidation of memories in the perirhinal cortex. The reviewers agreed that these findings were potentially interesting and important. However, the reviewers also felt that the manuscript suffered from a lack of clarity throughout all sections, including a lack of a clear setup of the rationale for the studies in the Introduction (with citations to relevant prior work). It is essential for the authors to improve the presentation of their Results to ensure that the paper can be read and understood by a broad audience. Most critically, reviewers also agreed that a key analysis is missing, an analysis that is needed to substantiate the study's conclusions. Specific comments are detailed below:

Essential revisions:

1) The authors' main conclusions are obscured by only reporting discrimination ratios during exploratory behavior. The interpretation of spontaneous exploration-based behaviors requires that raw values of exploration of incongruent and congruent objects be reported across all phases of the task. Specifically, the authors claim that the experimental manipulations after test 1 block the reconsolidation of the congruent object (A). If this is the case, then animals are 'forgetting' that they have experienced object A in the familiar context previously. This would be associated with enhanced exploration of object A. Thus, for example, during test 2 if the authors were to compare exploration of object A for rats that received emetine (EME) versus those rats that received vehicle, the EME group should have enhanced exploration relative to the vehicle control. Without a comparable analysis being performed, it is conceivable that the EME effect was due to this drug making the rat/mouse behave as if the novel object in test 2 was familiar. This possibility cannot be eliminated by presentation of the discrimination index alone. Another way to rule out this latter possibility would be to compare the exploration times of the novel objects in test 2 (C versus D) to test whether there was a difference in novel object exploration in EME versus vehicle conditions. Without such analyses, the main conclusions of the study are not well-founded.

2) The rationale for the studies require much better justification. It is unclear why certain studies were done and why others were not. Some restructuring of presentation could help. For example, the 'boundary conditions' experiments could be much better introduced and justified with regard to the processes that they address. In general, there should be a better lead-in to the circuits that will be examined.

3) Key references to relevant prior work are missing. There is no mention of Aggleton's work on object recognition, although his group has published many studies examining the role of the PRH and HPC that are relevant to this study. In addition, mentions to relevant work by Malcolm Brown should be expanded and elaborated.

4) There was a lack of a clear explanation of the behavioural/psychological mechanisms that support these data. A few specific examples are as follows. Why is the presentation of the incongruent object critical to induce reconsolidation of the congruent object in the PRH (i.e., is this related to prediction error?)? 5-HT2aR activity was not necessary to solve the OIC task when a comparison was made between a familiar OIC congruent object and a novel object, but was necessary between a familiar OIC congruent object and an incongruent OIC object. This suggests that the reported modulation of mPFC activity was specific to discrimination between a congruent and an incongruent OIC but not between a familiar (congruent) and novel object? Why would this be? Also, the idea of competing episodes is interesting, but it is unclear whether it would depend on the extent of similarity.

---

## [Author Response]

Essential revisions:1) The authors' main conclusions are obscured by only reporting discrimination ratios during exploratory behavior. The interpretation of spontaneous exploration-based behaviors requires that raw values of exploration of incongruent and congruent objects be reported across all phases of the task.

We believe that discrimination indexes give an intuitive measure, since they express an index of preference for the novel object. The discrimination index as we have calculated is standard in behavioral studies, and used in a number of different behavioral paradigms. However, we understand the reviewers' concerns regarding reporting only the index, as we could be misinterpreting the results. We have now added two additional tables as supplementary information that contain the raw values of the exploration times for the congruent and incongruent objects for the different experiments. We are reporting the data for each phase of the task. Supplementary files 1 and 2 contain the data of test 1 (total exploratory time and times for each object respectively). Supplementary file 3 contains the data of test 2.

Specifically, the authors claim that the experimental manipulations after test 1 block the reconsolidation of the congruent object (A). If this is the case, then animals are 'forgetting' that they have experienced object A in the familiar context previously. This would be associated with enhanced exploration of object A. Thus, for example, during test 2 if the authors were to compare exploration of object A for rats that received emetine (EME) versus those rats that received vehicle, the EME group should have enhanced exploration relative to the vehicle control. Without a comparable analysis being performed, it is conceivable that the EME effect was due to this drug making the rat/mouse behave as if the novel object in test 2 was familiar. This possibility cannot be eliminated by presentation of the discrimination index alone. Another way to rule out this latter possibility would be to compare the exploration times of the novel objects in test 2 (C versus D) to test whether there was a difference in novel object exploration in EME versus vehicle conditions. Without such analyses, the main conclusions of the study are not well-founded.

Thank you for this suggestion. As mentioned in response to the previous point, it would indeed be important to differentiate the different alternative explanations for our results. We run this analysis for the two main experiments of the manuscript described in Figure 1A and Figure 4A respectively. We added these analyses as a supplementary figure (Figure 1—figure supplement 2 and subsection “5-HT2aR activity in the mPFC modulates recognition memory reconsolidation in the PRH” and subsection “5-HT2aR blockade in the mPFC has no effect on reconsolidation of the OIC memory traces in the dorsal hippocampus”). As the reviewers predicted, the infusion of EME in the PRH after test 1 affected the exploratory time of the objects during test 2, supporting our view that EME is blocking the reconsolidation. In the first experiment in which EME was infused in the PRH, we see a change in the exploration of object A when we compared the EME vs. VEH groups. In the case in which EME was infused in the dHPC, we see differences in the exploratory time for object B (the one that reconsolidates in the dHPC). These results are consistent with what we reported based on the discrimination indexes.

2) The rationale for the studies require much better justification. It is unclear why certain studies were done and why others were not. Some restructuring of presentation could help. For example, the 'boundary conditions' experiments could be much better introduced and justified with regard to the processes that they address. In general, there should be a better lead-in to the circuits that will be examined.

Thank you for pointing this out and we apologize for not being clear enough. We made significant changes to the text to better address the rationale behind the experiments we performed. Regarding the boundary condition section we emphasized the rationale behind the experiments performed: i.e. since to our knowledge this is the first study analyzing reconsolidation of the OIC task, we wanted to control that some important hallmarks of the reconsolidation process were present in the OIC task as they were described for other object recognition task. We clarified this by adding a paragraph in this section (subsection “Boundary conditions for reconsolidation in the OIC task in PRH”)

3) Key references to relevant prior work are missing. There is no mention of Aggleton's work on object recognition, although his group has published many studies examining the role of the PRH and HPC that are relevant to this study. In addition, mentions to relevant work by Malcolm Brown should be expanded and elaborated.

Thank you very much for bringing this absence to our attention, we're sorry we forgot to include references to this important work. We have now cited relevant works from the Aggleton´s group as well as Brown´s group throughout the manuscript.

4) There was a lack of a clear explanation of the behavioural/psychological mechanisms that support these data. A few specific examples are as follows. Why is the presentation of the incongruent object critical to induce reconsolidation of the congruent object in the PRH (i.e., is this related to prediction error?)? 5-HT2aR activity was not necessary to solve the OIC task when a comparison was made between a familiar OIC congruent object and a novel object, but was necessary between a familiar OIC congruent object and an incongruent OIC object. This suggests that the reported modulation of mPFC activity was specific to discrimination between a congruent and an incongruent OIC but not between a familiar (congruent) and novel object? Why would this be? Also, the idea of competing episodes is interesting, but it is unclear whether it would depend on the extent of similarity.

Thank you for these comments. We apologize for not being comprehensible enough. We now realize that some of the concepts we were trying to present were not clearly presented. We modified the Discussion in order to introduce the topics suggested by the reviewers and to bring forward the main ideas we are trying to make with this manuscript. Briefly, we believe that the different patterns of reconsolidation in the PRH and the dHPC arise because this process is engaged by different types of mismatches. An object novelty engages reconsolidation in PRH, but a contextual mismatch does it for the HPC. In addition, 5HT2aR in the mPFC would be necessary for retrieval and reconsolidation when retrieval cues activate the representation of more than a single episode. Establishing the mechanisms by which 5HT2aR in the mPFC modulates this process will be the goal of future studies.